# Assessment of Building Energy Simulation Tools to Predict Heating and Cooling Energy Consumption at Early Design Stages

**Fernando Del Ama Gonzalo** [1,*] **, Belén Moreno Santamaría** [2] **and María Jesús Montero Burgos** [3]

1   Department of Sustainable Product Design and Architecture, Keene State College, 229 Main St., Keene, NH 03435, USA
2   Department of Construction and Architectural Technology, Technical School of Architecture of Madrid, Universidad Politécnica de Madrid, Av. Juan de Herrera, 4, 28040 Madrid, Spain
3   Facultad de Humanidades y Ciencias de la Comunicación, Campus de Moncloa, Universidad San Pablo-CEU, CEU Universities, 28040 Madrid, Spain
*   Correspondence: fernando.delama@keene.edu

**Abstract:** Recent developments in dynamic energy simulation tools enable the definition of energy performance in buildings at the design stage. However, there are deviations among building energy simulation (BES) tools due to the algorithms, calculation errors, implementation errors, non-identical inputs, and different weather data processing. This study aimed to analyze several building energy simulation tools modeling the same characteristic office cell and comparing the heating and cooling loads on a yearly, monthly, and hourly basis for the climates of Boston, USA, and Madrid, Spain. First, a general classification of tools was provided, from basic online tools with limited modeling capabilities and inputs to more advanced simulation engines. General-purpose engines, such as TRNSYS and IDA ICE, allow users to develop new mathematical models for disruptive materials. Special-purpose tools, such as EnergyPlus, work with predefined standard simulation problems and permit a high calculation speed. The process of reaching a good agreement between all tools required several iterations. After analyzing the differences between the outputs from different software tools, a cross-validation methodology was applied to assess the heating and cooling demand among tools. In this regard, a statistical analysis was used to evaluate the reliability of the simulations, and the deviation thresholds indicated by ASHRAE Guideline 14-2014 were used as a basis to identify results that suggested an acceptable level of disagreement among the outcomes of all models. This study highlighted that comparing only the yearly heating and cooling demand was not enough to find the deviations between the tools. In the annual analysis, the mean percentage error values showed a good agreement among the programs, with deviations ranging from 0.1% to 5.3% among the results from different software and the average values. The monthly load deviations calculated by the studied tools ranged between 12% and 20% in Madrid and 10% and 14% in Boston, which were still considered satisfactory. However, the hourly energy demand analysis showed normalized root mean square error values from 35% to 50%, which were far from acceptable standards.

**Keywords:** building energy simulation; cross-validation of software tools; energy efficiency in buildings

## 1. Introduction

Energy simulation procedures for buildings emerged from the oil crisis in the 1970s [1]. Over the previous decades, they have been implemented in various computer systems, taking advantage of their mathematical processing and graphical representation capabilities [2]. A variety of free or commercial building energy simulation software is currently available. For example, since 1996, the US Department of Energy has maintained a directory of approximately one hundred computational tools [3], in addition to developing some free ones. [4]. The US government developed the DOE-2 and BLAST building energy

analysis computer programs to provide energy simulation for commercial buildings in 1977. The US Department of Energy sponsored DOE-2, and Lawrence Berkeley Laboratory (LBL) was the lead laboratory in the DOE-2 development effort based mainly on ASHRAE algorithms. The BLAST building energy analysis computer program was developed by the US Army Construction Engineering Research Laboratory (CERL) with recent assistance from LBL [5]. EnergyPlus was a building performance simulation program that combined the capabilities and features from BLAST and DOE–2 along with new capabilities [6]. It was primarily a simulation engine with no user interface. DOE releases major updates to EnergyPlus twice annually, tested according to the ASHRAE Standard 140 methodology. An updated version 22.2.0, with bug fixes, was released on 30 September 2022. Several software tools are validated with ISO 13790:2008-09 and ASHRAE 90.1 standards [7,8]. Building energy simulation tools' differences depend on the mathematical models used to simulate the heat transfer between the external envelope and the indoor and outdoor environments. The main reasons to select the tool are the ease of use and accessibility of the source code. Thus, two typologies of BES tools are generally considered: general-purpose and special-purpose. The former allows users to define their mathematical models, making the tool more flexible with the drawback of the complication of use and a low calculation speed. The latter uses different predefined standard simulation algorithms and permits a high calculation speed, with the disadvantage of lower flexibility in the simulation of non-standard problems. TRNSYS and IDA ICE fall into the first typology, while Energy-Plus is in the second one [9]. TRNSYS, developed at the Solar Energy Laboratory at the University of Wisconsin-Madison, is a transient system simulation program for complex systems characterized by the division of a problem into a series of smaller subproblems contained in specific components or types. In addition, new models can be compiled into new parts and introduced in the TRNSYS library [10]. IDA ICE is a flexible whole-building performance simulation tool developed by the Division of Building Services Engineering, the Royal Institute of Technology in Stockholm (KTH), and the Swedish Institute of Applied Mathematics (ITM), which works with a program-independent language for modeling dynamical systems by using differential-algebraic equations [11]. EnergyPlus is a special-purpose tool developed by the US Department of Energy based on a modular structure with the possibility of adding validated new models [12]. Other programs have emerged due to legislation or specific certifications, such as LIDER-CALENER in Spain, CCTE in Chile, CEPE in California, and the Passive House Standard in Germany. Some apply steady-state conditions without changing parameters over time, and others consider transient conditions with yearly simulations. Some software tools are compatible, allowing the export of 3D digital models into energy simulation tools. Online tools are also available to analyze the energy performance of buildings, such as Autodesk's Green Building Studio or MIT's Design Advisor.

Building energy simulation (BES) software tools still need to be improved because energy performance concepts have been recently incorporated into academia and public or private building design processes [13]. According to some authors, the needs of architects and engineers in applying simulation tools are quite different [14]. The results showed that architects' needs were focused on design and that their most critical applications were summarized in shading calculation, passive heating, orientation, natural ventilation, and geometry [15]. Another conclusion is that the ease of use affects more than the accuracy of the results [16]. Four aspects have been considered to assess BES tools in the scientific literature: the graphic interface with the ability of the program to enter technical descriptions of materials and create customized and transferable libraries, the incorporation of climate data, the assistance provided by the software in both modeling and data entry, and the detail of the results output, formatting, and graphics [17]. Therefore, the impact of building energy simulation software varies according to the design stage in which it is applied [18]. Building designers can achieve more significant improvements at earlier stages, so architects are primarily responsible for most decisions that affect performance [19]. The implication of climate change has also been discussed. Simulations should predict

future behaviors based on records. Although it might be challenging to predict the changes expected from global warming in many regions of the planet, a moderate influence of climate change on the environmental behavior of buildings has been predicted [20].

Energy analysis or simulation tools include an interface with various features to calculate buildings' thermal performance and present the results [21]. The inputs are the site's climatic conditions, the building envelope's shape and materials, openings and surroundings, equipment, and occupancy patterns [22]. Based on this information, mathematical processes are conducted that indicate the interior environmental conditions (temperature, humidity, air quality, and $CO_2$ emissions) and energy demands over time. In addition, some characteristics of solar radiation, natural lighting, thermal conduction, air circulation, material life cycle, or environmental certification are also expressed by graphic or numerical visualization methods, which can be exported to other software tools. Some software benchmarks have been published [23,24]. These studies reveal similarities between the systems and highlight fundamental differences in the user interface and relationship with other software tools. A proposed methodology for validating the software identifies a standard performance in the calculations of thermal conduction, differences in convective aspects, and various uncertainties in the data entered [25,26]. In reviews of the same case using different software, up to a 30% disparity was detected in the results with the same indicators [27], attributed to specific technical considerations and local conditions. Other studies revealed differences in the outputs, mainly due to the complexity of the buildings [28]. Software evaluations generally focus on internal capabilities without reviewing implementation factors, such as costs, installation, support, or user training [29]. Therefore, there are still several relevant questions about the performance of commercial tools. The information resulting from the BES software can range from energy demands for different periods (annual, seasonal, or daily) to the range of comfort conditions. $CO_2$ emissions, energy flows, solar radiation, natural lighting, water consumption, and solar exposures of the fenestration at different times of the day or the year are also valuable outputs of energy simulation software [30,31].

The first section of this article studied the state-of-art simulation tools available in the market. The following sections addressed the definition of the conditions of the case study for comparison between different simulation and certification programs. Then, the factors that affect the energy simulations in general and, in particular, the terms of each program were studied. Finally, the results were discussed, and cross-validation was conducted with Design-Builder and IDA ICE as reference software tools that deliver daily and hourly energy consumption values.

## 2. Materials and Methods

There are different modeling strategies for the buildings. Simulation systems vary in their operations and the level of detail required, affecting the ease of use and the analysis. Software tools with high usability, but few capabilities can be considered "basic". These tools do not require high-performance computers or advanced knowledge of building science. Essential software tools and web-based services, such as Design Advisor and Green Building Studio, are characterized by their work speed and are usually free of charge. However, only a small number of aspects are analyzed. On the other hand, the programs with greater capacities, such as TRNSYS, IDA ICE, and Design-Builder, are characterized by their more significant level of detail in modeling, description of equipment, schedules, climatic files, diversity of analysis, verification of regulations, and relationship with other software. As a result, the outputs are more accurate but require high-performance equipment, advanced user knowledge, and more work time [32]. Building information modeling tools, such as Revit, recently combined a wide range of capabilities with a promising modeling interface well-known by architects, allowing for a comprehensive energy study of a home in a few hours [33].

## 2.1. Basic Simulation Software Tools

The basic programs offer predefined volumetric elements, indicating the main measurements and selecting general characteristics. Some authors stated that eQUEST helps make critical decisions for overall energy consumption, peak temperature prediction, and heating and cooling load calculations during the design phase. Its calculation engine is based on DOE-2, which has limitations in modeling emerging technologies. However, it is still used by professionals because it is quick to produce results and free of charge [34]. Design Advisor is an online software developed by the Massachusetts Institute of Technology (MIT) that calculates a building's energy demand (http://designadvisor.mit.edu/, accessed on 10 November 2022). In addition, it allows for a fast and intuitive evaluation of the lighting, ventilation, and life cycle conditions without having excellent knowledge of building physics. The modeling process is quite basic. For example, the user can only describe a rectangular area and select its dimensions. In the case of windows, the user can only work with a percentage of openings. The tool gives a range of values by default for non-expert users. Although these limitations make this phase fast and intuitive, accurate results are outside the scope of this tool. Users must select the climatic data from a limited number of cities in the US and the rest of the world. Unlike other basic simulation tools, Design Advisor allows users to enter the U-value of the envelope components instead of using the database that these software tools usually have. The setup screen allows the user to estimate how the natural lighting will be simply distributed in the model. It also measures natural ventilation levels. The strengths of this software are its friendly interface and ability to generate different scenarios of a model by entering the values of the materials. The results are presented clearly in graphic and written reports. In addition, the user can run up to four simulations and review all the details of the scenarios' performance, which makes Design Advisor a useful software for early design stages. The modeling system is very basic, so users can only analyze rectangular geometries, and the windows cannot be created only be considered as a percentage of the facade.

## 2.2. Advanced Simulation Software Tools

Some advanced programs import plans or geometries from other design programs or use free and essential modeling software. However, the imported information must be modified or complemented by dividing zones, establishing walls without thickness, or eliminating graphic details. Sometimes, generating the model in the energy simulation software is more effective. The study of advanced tools included EnergyPlus-based software, TRNSYS, and IDA ICE. The tools implement different models with different levels of detail to approach the numerical solution of the building system. An extensive description of the equations implemented in TRNSYS, IDA ICE, and EnergyPlus is provided in other articles [10]. As a general-purpose tool, TRNSYS can implement new mathematical models using programming languages, such as FORTRAN and C++. This ability makes TRNSYS an excellent asset for integrating disruptive technologies. However, due to its complexity, some authors do not recommend it for preliminary whole-building simulation [35]. Therefore, IDA ICE has been selected as a general-purpose tool because of its accurate import of BIM files. Some user interfaces for EnergyPlus have been developed over the last years with their interfaces, such as Sefaira Architecture, BuildSimHub, gEnergy, Design-Builder, and Cypetherm. Moreover, honeybee connects Grasshopper3D to EnergyPlus, and Insight and Green Building Studio are fully integrated with Revit. Finally, software development kits, such as Open Studio, are based on the EnergyPlus engine [36]. In this work, Design-Builder and IDA ICE are examples of special-purpose and general-purpose simulation tools due to the recommendation by the research community [37].

### 2.2.1. Design-Builder

Some typical features of Design-Builder are the calculation of the energy consumption of the building, evaluation of facade options in the event of overheating, and thermal simulation of buildings with natural ventilation. In addition, it can calculate the impact of supply air

distribution on temperature and velocity distribution within a room with computational fluid dynamics (CFD) [38]. In general, the program is quite intuitive, and the most significant complications that could arise are data entry. The results are easy to analyze since they deliver a large amount of data that can be selected according to what the user wants to visualize, comparing different graphs and giving the possibility to quickly assess the effects of the modifications made in the simulation. In the case of performing a computational fluid dynamics (CFD) analysis, the program allows the users to understand the parameters of thermal comfort and indoor air quality and to view the results in the same model [39,40]. Moreover, it even provides information not only on ventilation but also on all the calculations conducted by the program, significantly improving the understanding of how a building behaves. DB offers limited options for the development of models with complex geometries. Using EnergyPlus provides a detailed analysis of typical heating and cooling systems. Natural ventilation is considered with a set schedule so that night ventilation can be used as a saving strategy. Likewise, natural lighting systems and lighting control models calculate the savings in electric lighting. Finally, the software considers the shading by projections and blinds, interior panels, and exterior elements that could affect the lighting. The results break down the energy consumption by fuel type and end-use and display weather data and indoor temperature. Users can also analyze heating and cooling loads, infiltration, ventilation, and $CO_2$ generation. Parametric analysis screens allow the user to investigate the effect of variations in design parameters on several performance criteria [41].

### 2.2.2. IDA Indoor Climate Energy

IDA Indoor Climate and Energy (IDA ICE) is a software for building simulations of energy consumption, indoor air quality, and comfort. It covers thermal models, $CO_2$ and moisture calculations, and room temperature gradients. The software's ultimate goal is to be capable of providing valuable information at all stages of a building's design. It allows dynamic thermal simulation, natural and artificial lighting evaluation, calculating thermal loads, and simulation of systems based on components. IDA ICE also provides data for studying the indoor thermal climate and the energy consumption of the entire building. This software interface is designed to create and simulate simple and advanced cases. Moreover, it imports BIM files with accurate geometries. As a result, IDA ICE can conduct energy studies and complete designs, including the envelope, systems, plant, and control systems. The methods used by IDA ICE facilitate the development of the software to adapt it to the requirements and local languages and extend it with new modeling capabilities.

### 2.3. Building Information Modeling Tools

Building information modeling (BIM) software allows architects to model a three-dimensional building so that designers can use the information for other goals, such as predicting energy consumption. BIM represents the building as a database that integrates aesthetic aspects and thermal properties with all information in the building [42]. As a result, BIM simplifies thermal analysis and offers results to reduce energy needs and analyze renewable energy options. BIM tools can also analyze building form, optimize the building envelope, conduct daylighting analysis, and integrate renewable energies [43]. In this article, all building elements were modeled using Revit, and energy settings were used to modify the building, location, and detailed data about ventilation rate and mechanical equipment. Then, this information was forwarded to Green Building Studio for energy simulation. Autodesk Green Building Studio (GBS) is a web-based service that includes building energy and carbon analysis tools. GBS calculates yearly energy consumption, peak demand for electrical energy, life cycle analysis of energy and costs, generation of photovoltaic power, analysis of water use, calculations of potential natural ventilation, and carbon emissions. Furthermore, GBS can estimate renewable energy production through solar panels and wind [44]. In addition, BIM helps architects and engineers select optimal design strategies to reduce carbon emissions, energy, and cost via optimization models [45].

*2.4. Case Study*

A case study was used to compare the energy simulation programs, representing the characteristics of the office building typology. Office spaces usually combine a simple geometry with demanding internal loads and solar heat gains. The former makes office spaces a good case study for tools with simple 3D modeling capabilities, such as Design Advisor. The latter allows for studying the simulation characteristics of advanced energy simulation tools, such as Design-Builder and IDA ICE. Hence, the selected case study combines an open workspace for more than ten people with semi-enclosed workspaces for two to six people suitable for teamwork. Furthermore, ventilation and artificial lighting requirements are more relevant in office buildings than in other typologies. In addition, the high window-to-wall ratio makes this typology more demanding on cooling loads, even during mild weather seasons [46]. The case study consisted of a small isolated rectangular office building with a floor area of 160 m$^2$, brick masonry walls, concrete floor on the ground, without a crawl space, triple glass windows, aluminum frames with a thermal bridge break, a roof structure with a concrete slab and light insulation. The case study was tested in two locations, the city of Boston, US (42.35866° N, −71.05674° E), and Madrid, Spain (40.42028° N, −3.70577° E). Boston has a humid continental and subtropical climate, very common on the southern coast of New England. The hottest month is July, with an average maximum temperature of 28 °C and a minimum of 19 °C, humid conditions. On the other hand, the coldest month is January, with maximum average temperatures of 2 °C and a minimum of −6 °C. Temperatures exceeding 32 °C in summer and −12 °C in winter are unusual. Madrid's weather is dry, without too much rainfall throughout the year. Because Madrid is 650 m above sea level, this city experiences very different temperatures in the summer and winter. The warmest and most pleasant weather happens between May and mid-July when temperatures average between 20 °C and 32 °C. Towards the end of July and throughout August, it can get quite hot, sometimes reaching 40 °C. Nighttime temperatures remain around 18 °C. September is a pleasant month, with temperatures dropping to around 25 °C. October is still quite warm, with an average daytime temperature of 20°C. Figure 1 illustrates the architectural 3D model created with Autodesk Revit and the energy management system used for the thermal simulation. An air-handling unit with heating and cooling coils connected to a gas boiler and an air-cooled chiller provides the spaces with ventilation. The air-handling unit did not have a heat recovery, so if heating and cooling loads exceeded the capabilities of the air-handling unit, there were radiators as zone heat sources and indoor split units as extra zone cooling. Natural gas was considered the primary heating source, whereas electricity was the primary cooling source.

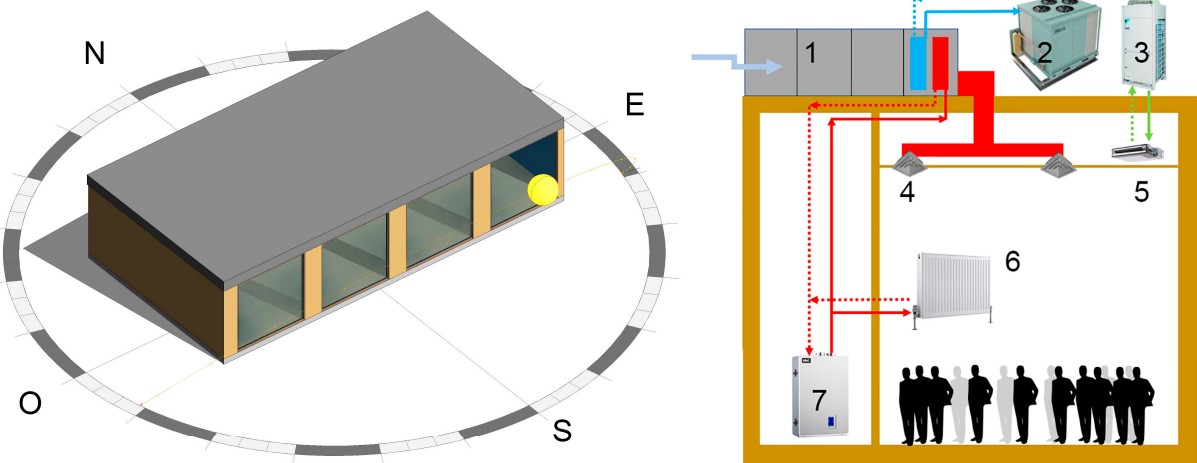

**Figure 1.** 3D model created with Autodesk Revit and the energy management system simulated by all the software tools. 1. air handling unit; 2. air-cooled chiller; 3. heat pump; 4. supply air diffusers; 5. indoor split unit; 6. radiator; 7. gas boiler.

The first step to simulate the same building in different software tools was the introduction of the building geometry. Then, the outdoor and indoor conditions were introduced using the default options of each tool. Finally, the calculation methods for heat transmission in the opaque and transparent parts of the envelope and the aspects that affected the ventilation and air quality were defined. The geometry of the case study was first modeled using Revit. It is a widely used program for architects due to its friendly interface and powerful modeling options. However, for the subsequent use of the geometry defined in the energy calculation software, it is necessary to follow specific definition rules that are not intuitive for the inexperienced user. Expert CAD users can create 3D models of the building, although the energy analysis requires a simplified model free from complicated families or textures that affect only the aesthetics. The next step was to export the 3D BIM model to other tools. Design Advisor worked only with elementary geometry and was allowed to simulate rectangular rooms. IDA ICE read the BIM file exported as a *.ifc extension. The IDA ICE geometry was entirely accurate when the BIM model was simplified. Although Design-Builder can import BIM files, the result was not satisfactory, and the Design-Builder interface was used to create the 3D model.

The default weather data for each tool was used. Design Advisor uses hourly climate data in TMY2 format for limited locations and cities. The weather file includes outdoor dry bulb temperature, outdoor relative humidity, direct and diffuse incoming solar radiation, incoming solar illuminance, and wind data with velocity and direction. IDA ICE handles two types of weather data: design days and yearly weather files. Design days consider the daily extreme wet and dry temperatures. Weather files are stored in the standard text format of the IDA time series (*.PRN) with a time resolution that can be set at hourly measurements. ICE is delivered with a separate utility program to convert some of the established weather data file formats into PRN files. When working with EnergyPlus as a calculation program, the climatic files are in *.epw format, containing all the normative and non-normative climatic variables. This format collects the hourly value of various climatic parameters for an average year. The published data must be converted from solar time to local time, which implies data interpolation and incorporating parameters other than the basic ones required by the simulation programs. For example, the climatic zone to which the building site belongs is selected in the program interface. In addition, it is possible to choose a different and personalized climatic file. Revit and Design-Builder shared the same EnergyPlus engine, so they used the same climate file.

Regarding indoor conditions, three aspects were analyzed: the use of the building, the internal loads, and the effect of solar radiation through windows that was reflected by the interior elements. In all the analyzed tools, different uses can be selected through a drop-down menu when analyzing commercial buildings in the analyzed tools. The building use and the schedule affected the differentiated heating, cooling, and ventilation conditions. The "office building" type was selected to conduct an adjusted comparison with the set of buildings in the database. Shadow programming and other aspects characterized the structure's dynamic analysis, such as the radiant and convective fractions of the different surfaces. The total power per area associated with internal loads was 100 W per person. Consequently, the total occupancy was 38 people, with a total floor area of 158 $m^2$ and a total power density per area of 4.4 ($W/m^2$). The lighting and equipment power were associated with the same schedule and had a radiant ratio of 0.656 for the lighting and 0.7 for the equipment. When it came to solar heat gains, Design Advisor and Revit considered only direct solar radiation, meaning they were not modeling the radiant exchanges of the adjacent surfaces of the building other than the ground with a reflectivity of 0.2 by default. IDA ICE and Design-Builder considered the exterior finishes of the building envelope and the radiant exchanges with the outside. The default solar reflectance coefficient for opaque elements was 0.6 for exterior walls, 0.15 for roofs, and 0.5 for interior partitions, whereas the emissivity for exterior walls and roofs was 0.9.

The opaque envelope elements were defined using the Revit element description. Design-Builder and IDA ICE allowed the enclosures to be defined layer by layer or to

use a simplified definition, whereas, in Design Advisor, the only parameter that could be introduced was the thermal resistance of the elements. Internal thermal mass in Design Advisor was defined by adding a floor slab with an equivalent mass to specify the items within the space that were important to heat transfer calculations. Thus, the thermal transmittance of the envelope elements was 0.35 $(W/m^2K)$ for walls and roof, 0.31 $(W/m^2K)$ for floors, and 1.6 $(W/m^2K)$ for interior partitions. Heat transmission in transparent and translucent openings was defined similarly to opaque enclosures. The determined properties did not vary with the sun's angle of incidence, so they were fixed parameters. The glazing panes were considered materials without mass and, therefore, without inertia. However, they did have the constructive characteristics recommended by the regulations, with a thermal transmittance of 1.42 $(W/m^2K)$ and a solar heat gain coefficient of 0.44. Finally, the ventilation of the building and the permeability of the doors and windows were defined. The building energy model defined ventilation with a constant nominal value of 2.5 ACH. Since some of the analyzed tools did not have inputs for heat recovery devices, no heat recovery was considered for the simulation. The infiltration rate was set to 0.72 air changes per hour (ACH). In Design Advisor, the defined infiltrations are of a constant type and not dependent on the outside temperature or the wind speed. On the other hand, design-Builder and IDA ICE consider wind speed's influence on infiltrations.

## 3. Results

This section shows a breakdown of the specific characteristics of the analyzed advanced software tools, EnergyPlus and IDA ICE. Finally, the energy demand of the selected buildings through all the software tools will be shown and discussed. Commercial software tools have several common hypotheses in modeling energy simulation. Specific parameters are modeled, and some are directly set by default. The same case study has been tested in each simulation tool. However, the inputs were different due to the operating principles of each software.

### 3.1. Design Advisor

The Design Advisor output provides a straightforward path for viewing and comparing the energy performance of the case study. When the simulation process is concluded, visual and numerical results for energy consumption, life cycle costs, comfort, and daylighting analysis are displayed as a pdf report. The energy model calculates the energy required for the building's heating, cooling, and lighting. The results include the outdoor temperature and consider direct and diffuse thermal and visible solar radiation. The internal heat gain considers the sum of the occupants, equipment, and lighting loads, depending on the schedule introduced as an input. The thermal mass is considered as a concrete slab on the room's total floor area. The solar thermal energy that enters through the windows is assumed to be absorbed by the floor slab. Finally, the number of people and the schedule define the ventilation loads. The results are displayed monthly and annually with the capability of viewing four simulations simultaneously for performance comparisons. Figure 2 shows the energy consumption in Wh per $m^2$ of the case study in two different locations, Madrid and Boston. The heating and cooling energy loads differ from the Boston case to the Madrid case due to the differences in weather conditions. Figure 2 illustrates the energy required to meet the heating and cooling loads. The software assumes a perfect conversion of fuel into heat energy in the boiler, whereas the coefficient of performance (COP) for cooling is 3.0. The most significant difference between Boston and Madrid occurred in the peak heating and cooling energy values. The peak heating energy in Madrid was 58 $(Wh/m^2)$, whereas, in Boston, it was 80 $(Wh/m^2)$. As it was expected, the cooling load in Madrid reached a peak of 59 $(Wh/m^2)$. The maximum Boston cooling load was 40.4 $(Wh/m^2)$.

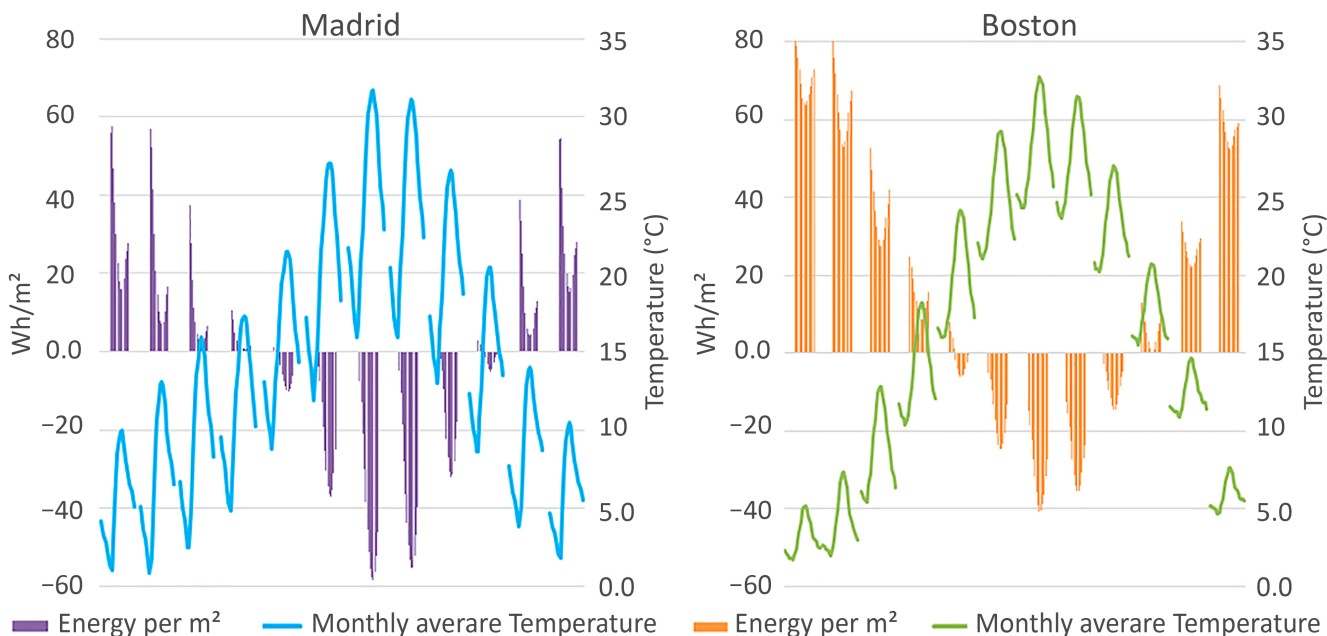

**Figure 2.** Results from Design Advisor. Energy per unit area and outdoor temperature for both locations.

### 3.2. Revit and Green Building Studio (GBS)

The temperature difference kept increasing until a peak value of 6°C at 7:00 p.m. The Revit and GBS presented limitations in the input of weather data. For example, the user cannot update or upload weather data from other sources as is possible in EnergyPlus. Calculating energy performance in the case study with different locations provides the users with building performance factors, such as energy use intensity, renewable energy potential, annual carbon emissions, and energy usage (fuel and electricity). Monthly heating and cooling loads, fuel, and electricity consumption are other outputs from the software energy analysis. GBS and Revit cannot calculate the internal temperature, only the heating and cooling loads, so these tools are not used to study thermal comfort. Figures 3 and 4 depict the peak heating and cooling power in both locations broken down into two places, the offices and the lobby. From the analysis of the simulation results shown in Figure 3, it was concluded that the peak heating load in Boston was 17,500 W, whereas the peak cooling load was 27,850 W.

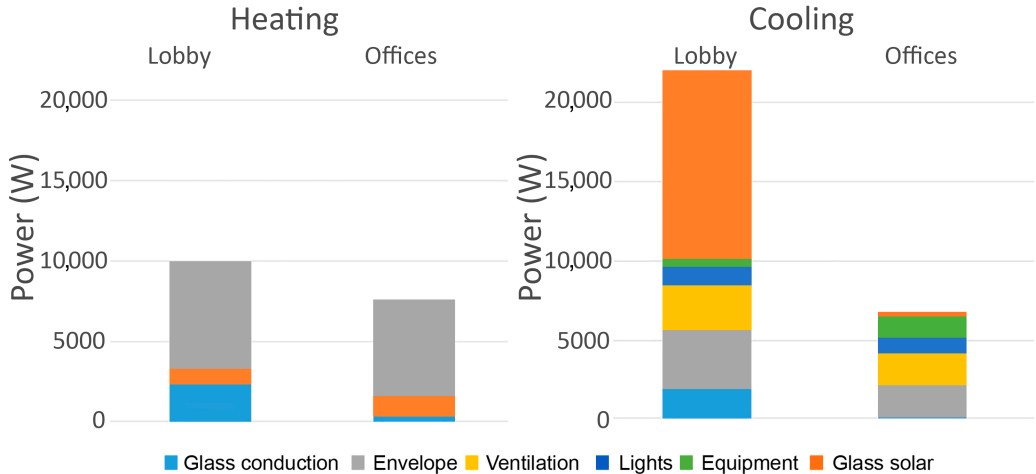

**Figure 3.** Results from Revit. Peak heating and cooling power in the Boston case study.

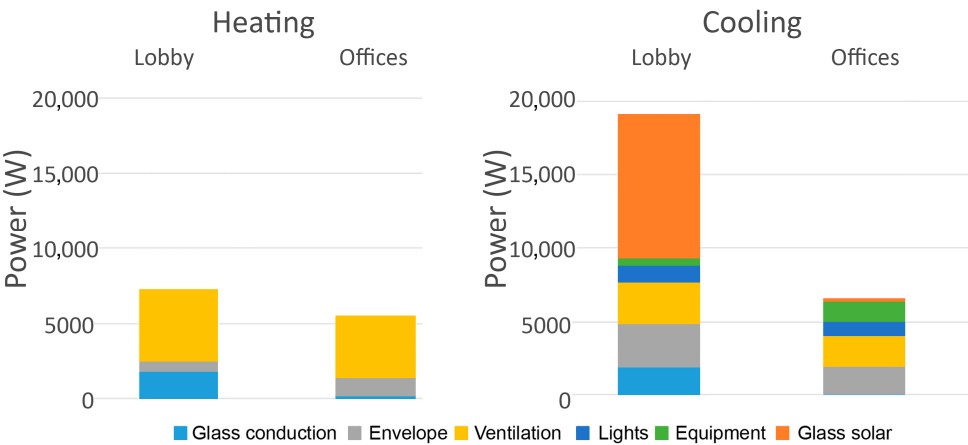

**Figure 4.** Results from Revit. Peak heating and cooling power in the Madrid case study.

Figure 4 shows the heating loads in Madrid with peaks of 25,670 W and 12,030 W for cooling and heating, respectively.

### 3.3. Design-Builder with EnergyPlus

Design-Builder's outputs include various parameters of the building and present them to the user graphically or through tables. It illustrates hourly, monthly, and yearly data with graphs that can be easily edited, which makes it straightforward to analyze the influence of different energy management strategies. Figure 5 illustrates the simulated heating and cooling loads in kWh. Using custom versus default inputs, including operating schedules, can make differences in the simulation outputs, so it is essential to go over all the default data and change them according to the characteristics of the project. Given the results in Figure 5, analyzing the hourly building's internal loads was necessary to understand the influence of the parameters, such as the default occupancy, plug-load, and lighting settings.

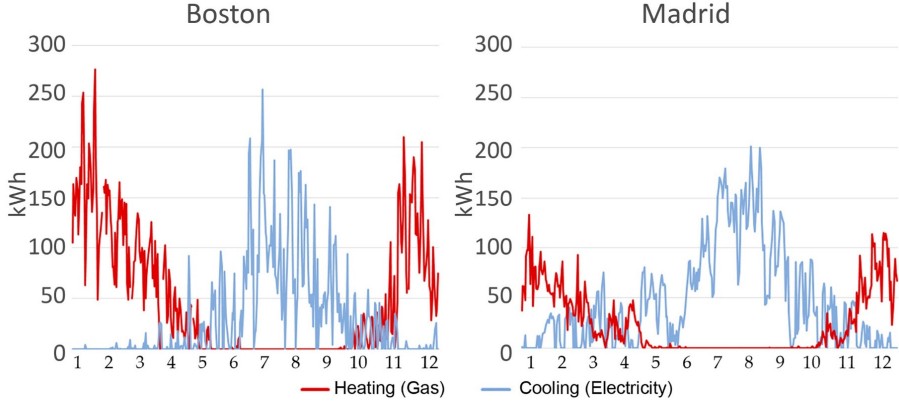

**Figure 5.** Results from Design-Builder. In both locations, annual heating and cooling energy demand in kWh of the case study.

The conversion factors of fuel (natural gas) into heat energy and electricity were considered 1.1 and 1.954, respectively, whereas the coefficient of performance (COP) for cooling was 3.0. Annual heating consumption was 19,568 kWh in Boston and 8495 kWh in Madrid. The yearly cooling energy needed in Madrid, 17,160 kWh, was far more important than the cooling energy required in Boston at 12,001 kWh. Therefore, this result emphasized the advantages of carefully reviewing default parameters.

### 3.4. IDA ICE

The IDA ICE engine calculates the indoor conditions using a simplified zone model for energy simulations. It has been used to compare the power levels for primary and

secondary energy system components and the total energy consumption throughout the year. In addition, the model calculates the heat transfer coefficients. IDA ICE produces output files with different time resolutions. Hourly, daily, weekly, or monthly averages are shown as charts or tabulated. A particular function enables the export of the time series as a spreadsheet. Figure 6 shows the monthly performance of the case study in Boston and Madrid. The energy management system included an air-handling unit (AHU) without a heat recovery for ventilation. The AHU had heating and cooling coils. However, it was necessary to supply more energy with a secondary energy system inside the rooms. The input data led to a monthly peak load of 10,210 W for heating in January and 10,182 W for cooling in July.

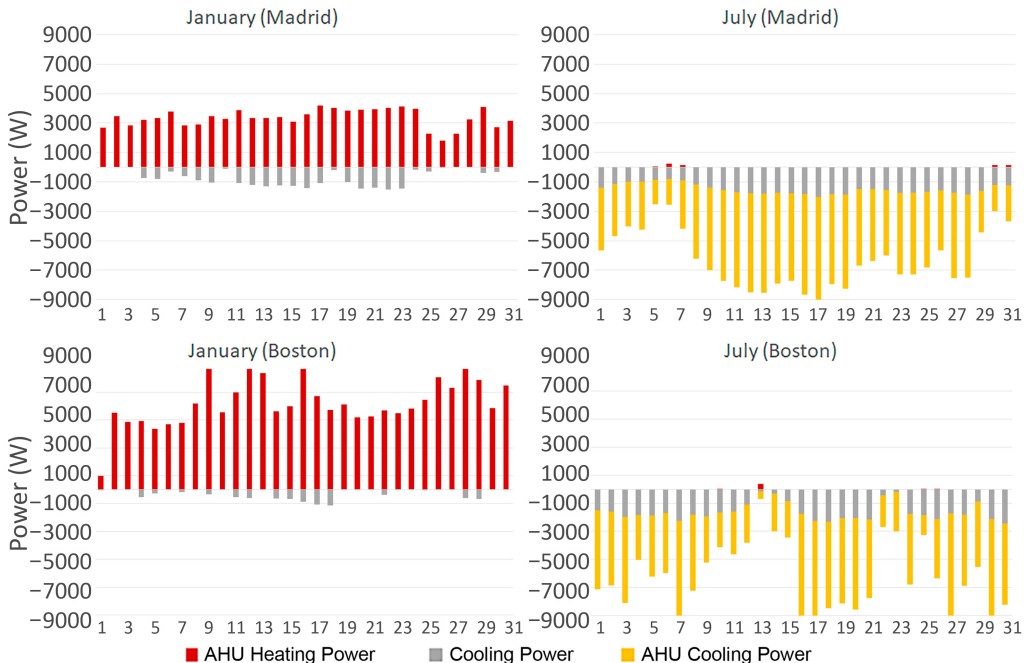

**Figure 6.** Results from IDA ICE. Heating and cooling power of the case studies in January and July.

In Madrid, the peak heating power was 5190 W in January, whereas the peak cooling power was 9077 W in July.

## 4. Discussion

Building energy simulation tools still need to improve their compatibility with building information modeling. For example, Design-Builder could have shown a better performance when importing the geometry from BIM software, such as Revit. Instead, the architect had to use its interface to model a simplified and inaccurate geometry. IDA ICE supported IFC models generated by Revit, although the Revit file needed to be simplified from external elements that hindered the energy simulation. Although the case study was simple and well described, multiple iterations were required to achieve a good agreement among the results from different tools.

According to the scientific literature, there are several methods for validating building energy simulation tools [34]: empirical validation—in which calculated results are compared with measured data from an actual building or test cell; analytical verification—in which outputs from software are compared with results from a known analytical solution or a generally accepted numerical method; and cross-validation—in which a validated program is compared with other validated programs. This article utilized the latter method to compare the results from different software tools. The 3D model was initially created using Autodesk Revit, so the first analysis must include operability between BIM files and the two advanced simulation tools, IDA ICE and Design-Builder. When it came to geometry

and materials, the overall geometry and orientation of the case study were successfully exported from Revit to IDA ICE. Spaces created in Revit could be used in the IDA ICE interface. However, the model must be simplified for issues with the plenum spaces and thermal zones. In addition, neither Revit nor the IDA ICE tools provided detailed instructions for any technique for confirming that the geometry was inaccurate. Design-Builder could not import the geometry accurately, so the authors created a 3D model using its interface and templates to assign the thermal parameters of the envelope and the HVAC systems. The cross-validation process required many iterations until the deviations among the studied tools were acceptable. The first iteration considered the default values required by the Design Advisor tool, such as the thermal transmittance of opaque and transparent envelopes, the solar heat gain coefficient of windows, the occupancy and ventilation rate, and the size of the building. The thermal mass was set as high, and heat recovery devices were not considered. After the first analysis, there were deviations in the yearly heating and cooling loads. The most significant difference among the results was observed in Madrid's cooling consumption. IDA ICE showed the highest cooling values in August with 5183 kWh, whereas Design-Builder's result was 3487 kWh. The second iteration was related to the total power density per area, the lighting power level, fixed at 5.4 (W/m2), and a building schedule of 91 h per week. Finally, the reflectance values were matched between Design-Builder and IDA ICE with values of 0.6 and 0.15 for exterior and interior finishes, respectively. Additionally, some inputs were different, and some parameters needed to be better considered when introducing data. The construction materials and thermal parameters were generally correctly imported for Design-Builder and IDA ICE software tools. However, they did not recognize layers with zero thickness, such as waterproof membranes. In terms of energy parameters and mechanical systems, it was necessary to reassign all the energy data according to the tool's HVAC libraries based on the imported space types. The thermal parameters of each room had to be specified again by the authors. The occupancy in Revit can be changed under the space properties, including the lighting and power schedule. Therefore, these operational schedule data had to be reassigned to the two advanced software tools, allowing more flexibility than Revit and affecting the energy analysis results, but with a lesser effect on overall energy consumption. The HVAC options are very restricted in Revit, as only one HVAC system can be assigned to the whole building. This limitation, although acceptable at early design stages, can lead to errors as the complexity of the building increases. Thus, all the systems were again defined in Design-Builder and IDA ICE to validate the energy analysis outcomes.

### 4.1. Yearly and Monthly Analysis

In the present article, a detailed cross-comparison was conducted, focusing on a case study that includes shading in southern windows. The energy demand was compared monthly and hourly for one year of simulation. The results presentation and the detail level (yearly, monthly, or daily) can influence decision-making. For example, graphs and diagrams help evaluate simulation programs, but objective data is needed to compare the case study's performance. Hence, the energy demand per unit area was used to assess the tools. The energy efficiency performance index (EEPi) is the energy intensity calculated as the yearly energy consumption by area. Equation (1) expresses the parameter that assesses building performance in kWh/m$^2$-year [23].

$$EEPi = \frac{Annual\ energy\ use}{Conditioned\ area} . \tag{1}$$

The results of the simulations did not show significant differences in the winter. The heating simulation tools gave similar results in both locations, Madrid and Boston, whereas the differences in cooling loads were much more remarkable. Figure 7 shows the results of the Boston case study simulated with all the studied tools. All simulation results showed that heating was far more important than cooling.

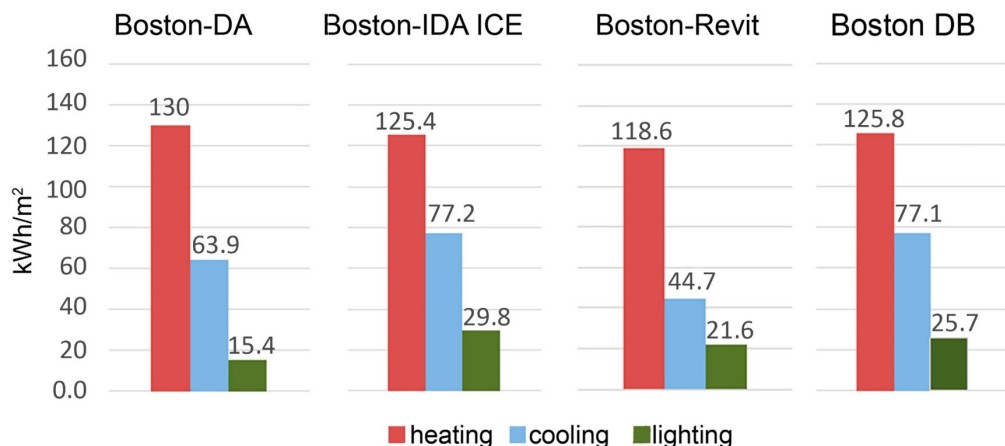

**Figure 7.** Yearly lighting, heating, and cooling demand for the climate of Boston for all the tools.

The highest heating energy consumption reached 130 (kWh/m$^2$) with the Design Advisor tool, and the lowest was 118.6 (kWh/m$^2$) with Revit. When it comes to cooling loads, the maximum yearly energy per unit area was 77.2 (kWh/m$^2$) with IDA ICE, whereas the minimum was 44.7 (kWh/m$^2$) with Revit. Figure 8 illustrates the yearly kWh per unit area for the Madrid case study simulated with all the tools. The highest cooling energy consumption reached 114.5 (kWh/m$^2$) with Revit, and the lowest was 107.3 (kWh/m$^2$) with Design-Builder. When it came to heating loads, the maximum yearly energy per unit area was 61.9 (kWh/m$^2$) with Design Advisor, whereas the minimum was 50.0 (kWh/m$^2$) with Revit.

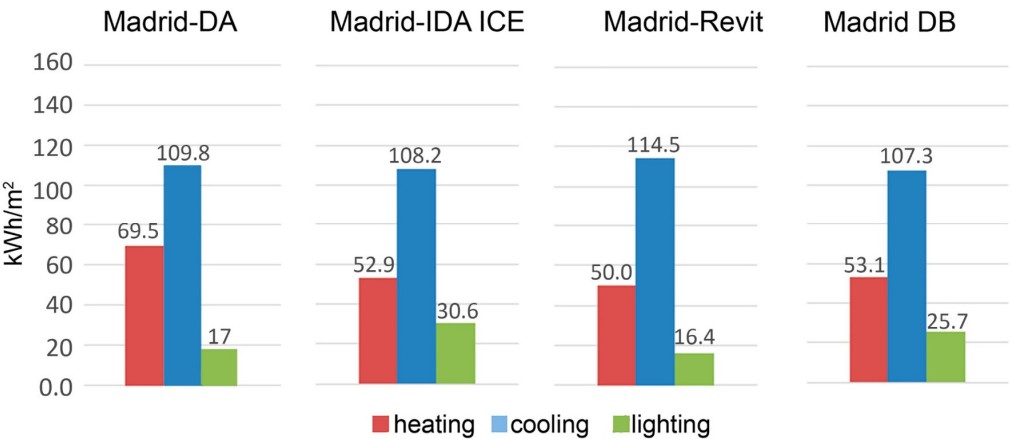

**Figure 8.** Yearly lighting, heating, and cooling demand for the climate of Madrid for all the tools.

As was expected in a building with no solar protection, the cooling loads in Madrid were more relevant than the heating loads. Figure 9 illustrates the monthly energy demand of the case studies simulated with three tools, Design Advisor, IDA ICE, and Design-Builder. The most significant difference among the results can be observed in the cooling consumption in Madrid. Design Advisor showed the highest cooling values in July with 4688 kWh, whereas Design-Builder´s result was 3856 kWh. However, the energy consumption in March and April with Design-Builder was much higher than that of IDA ICE. The use of statistical indices quantified the deviations between the outputs of the simulation tools.

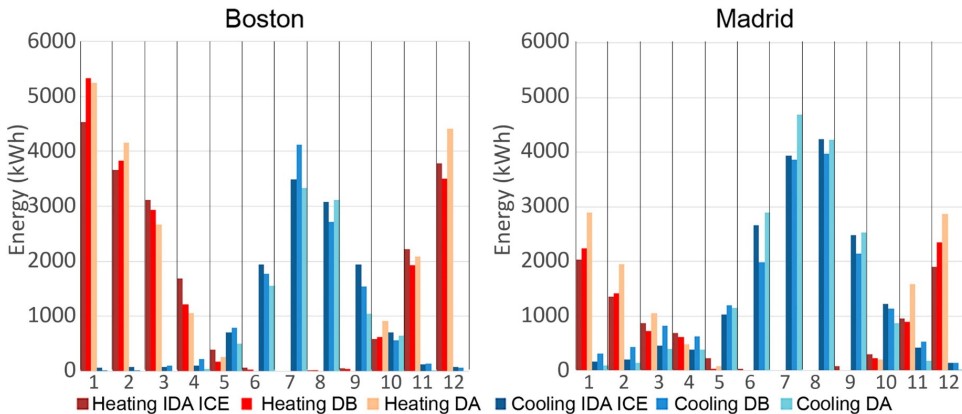

**Figure 9.** Monthly energy demand of the case studies simulated with three different tools.

The monthly average value of each parameter was used as a reference to compare the software tools. Table 1 shows monthly values for heating and cooling and the average for the case study in Boston, whereas Table 2 shows the same values for the Madrid location. The mean error (ME) is the difference between the reference value and simulation results. The mean percentage error (MPE) is the average of percentage errors by which the reference values differ from the simulated values of the cooling and heating energy. Both expressions are shown in Equations (2) and (3).

$$ME = \frac{1}{n} \sum_{i=1}^{n} |S_{Si} - A_{Ri}|, \tag{2}$$

$$MPE = \frac{1}{n} \sum_{i=1}^{n} \left| \frac{S_{Si} - A_{Ri}}{S_{Si}} \right| 100, \tag{3}$$

where $S_{Si}$ is the simulated value, $A_{Ri}$ is the average value taken as a reference, and $n$ is the number of samples.

**Table 1.** Energy demand in Boston throughout the year with average values.

| | Heating (kWh) | | | | Cooling (kWh) | | | |
|---|---|---|---|---|---|---|---|---|
| **Month** | **IDA** | **DB** | **DA** | **Avg** | **IDA** | **DB** | **DA** | **Avg** |
| 1 | 4532 | 5329 | 5248 | 5036 | 55 | 2 | 0 | 19 |
| 2 | 3656 | 3829 | 4160 | 3882 | 71 | 12 | 0 | 28 |
| 3 | 3112 | 2927 | 2672 | 2904 | 79 | 97 | 0 | 59 |
| 4 | 1688 | 1212 | 1056 | 1319 | 102 | 221 | 32 | 118 |
| 5 | 384 | 165 | 256 | 268 | 706 | 787 | 496 | 663 |
| 6 | 59 | 29 | 0 | 29 | 1934 | 1764 | 1552 | 1750 |
| 7 | 0 | 0 | 0 | 0 | 3486 | 4124 | 3328 | 3646 |
| 8 | 2 | 0 | 0 | 1 | 3079 | 2710 | 3120 | 2970 |
| 9 | 55 | 37 | 0 | 31 | 1942 | 1536 | 1040 | 1506 |
| 10 | 579 | 619 | 912 | 703 | 706 | 556 | 640 | 634 |
| 11 | 2218 | 1923 | 2080 | 2074 | 120 | 134 | 16 | 90 |
| 12 | 3786 | 3498 | 4416 | 3900 | 72 | 57 | 0 | 43 |
| Total | 20,071 | 19,569 | 20,800 | 20,147 | 12,351 | 12,001 | 10,224 | 11,525 |

**Table 2.** Energy demand in Madrid throughout the year with average values.

| | Heating (kWh) | | | | Cooling (kWh) | | | |
|---|---|---|---|---|---|---|---|---|
| Month | IDA | DB | DA | Avg | IDA | DB | DA | Avg |
| 1 | 2030 | 2237 | 2896 | 2387 | 162 | 314 | 96 | 191 |
| 2 | 1351 | 1408 | 1952 | 1570 | 197 | 433 | 144 | 258 |
| 3 | 870 | 727 | 1056 | 884 | 459 | 826 | 400 | 562 |
| 4 | 687 | 620 | 480 | 596 | 380 | 624 | 384 | 463 |
| 5 | 232 | 39 | 80 | 117 | 1029 | 1195 | 1152 | 1125 |
| 6 | 35 | 5 | 0 | 14 | 2657 | 1989 | 2896 | 2514 |
| 7 | 12 | 0 | 0 | 4 | 3933 | 3856 | 4688 | 4159 |
| 8 | 3 | 0 | 0 | 1 | 4237 | 3975 | 4224 | 4145 |
| 9 | 87 | 4 | 0 | 30 | 2484 | 2143 | 2528 | 2385 |
| 10 | 298 | 223 | 208 | 243 | 1219 | 1140 | 864 | 1074 |
| 11 | 953 | 889 | 1584 | 1142 | 422 | 526 | 176 | 375 |
| 12 | 1900 | 2344 | 2864 | 2369 | 137 | 139 | 16 | 97 |
| Total | 8458 | 8496 | 11,120 | 9358 | 17,315 | 17,160 | 17,568 | 17,348 |

By computing the MPEs, the minor percent errors indicate that they are close to the average value.

Regarding the annual values, the results in Tables 1 and 2 indicated that the models performed well relative to one another. The annual space heating demand in Boston for the case study is estimated to be 20,071, 19,569, and 20,800 kWh by IDA ICE, Design-Builder, and Design Advisor, respectively, giving an average of 20147 kWh. Design Advisor showed the highest values for heating in both locations. In Boston, the heating absolute deviations of 729 kWh and 1231 kWh from IDA ICE and Design-Builder, respectively. In Madrid, the heating deviations were even higher, with 2662 kWh from IDA ICE and 2624 kWh from Design-Builder. Although monthly deviations were more relevant among the tools, the three models performed surprisingly well in an absolute sense. For example, the most significant deviation was in January heating values in Madrid with 2030, 2237, and 2896 kWh by IDA ICE, Design-Builder, and Design Advisor, respectively. IDA ICE showed the largest value for cooling loads in Boston with 4124 kWh, whereas IDA ICE and Design Advisor showed 3486 and 3328 kWh, respectively. The dataset used for this analysis was too small to determine the accuracy of the models, so an hourly analysis was required to understand the daily energy consumption.

In addition to the model's ability to predict yearly and monthly energy use, Tables 3 and 4 showed the Mean Absolute Error and the Mean Percentage Error of yearly and monthly predictions. The results showed that for this dataset, all models performed well. Table 3 illustrates the yearly energy demand analysis. The deviation of the results from all software tools showed values ranging from 0.1% to 5.3%, which are considered within the acceptable error range for MPE in building energy analysis. Some authors stated that acceptable errors could range between 5% and 15% [47]. The MPE analysis risk consisted of some cases where the simulated value was zero or very close to zero.

**Table 3.** Mean Error and Mean Percentage Error for yearly results from IDA ICE, Design-Builder, and Design Advisor for both case studies.

| | Madrid Heating | | | Madrid Cooling | | | Boston Heating | | | Boston Cooling | | |
|---|---|---|---|---|---|---|---|---|---|---|---|---|
| | IDA | DB | DA | IDA | DB | DA | IDA | DB | DA | IDA | DB | DA |
| kWh/m$^2$ | 52.8 | 53.1 | 69.5 | 108.2 | 107.2 | 109.8 | 125.4 | 125.8 | 130.0 | 77.1 | 77.1 | 63.9 |
| Avg. | | 58.5 | | | 108.4 | | | 127.1 | | | 72.7 | |
| ME | 5.6 | 5.4 | 11.0 | 0.2 | 1.2 | 1.4 | 1.6 | 1.3 | 2.9 | 4.5 | 4.4 | 8.8 |
| MPE(%) | 3.5 | 3.4 | 5.3 | 0.1 | 0.4 | 0.4 | 0.4 | 0.3 | 0.7 | 1.9 | 1.9 | 4.6 |

**Table 4.** Mean Error and Mean Percentage Error for monthly results from IDA ICE, Design-Builder, and Design Advisor for both case studies.

| | Madrid Heating | | | Madrid Cooling | | | Boston Heating | | | Boston Cooling | | |
|---|---|---|---|---|---|---|---|---|---|---|---|---|
| | IDA | DB | DA | IDA | DB | DA | IDA | DB | DA | IDA | DB | DA |
| ME | 133 | 76 | 186 | 97 | 191 | 175 | 155 | 102 | 149 | 98 | 101 | 134 |
| MPE (%) | 20 | 12 | 20 | 14 | 21 | 37 | 13 | 14 | 10 | 24 | 20 | 18 |

The proposed monthly energy demand analysis did not consider energy values below 100 kWh per month, so the number of samples, *n*, was reduced accordingly. Table 4 shows the ME and MPE for all the case studies. The minimum MPE of 7% can be observed for heating conditions. The maximum deviation of 37% from the average value occurred in the Madrid cooling analysis with the software Design Advisor. The rest of the deviations are consistent with values around 15%, so the results from this data set were usually within the acceptable margin of error.

Table 3 showed that the model's yearly MPE was below 6% in all cases, whereas Table 4 showed the values of error calculated by each software for each location. The results showed that the MPE of the monthly space heating loads calculated by the different tools ranged between 12% and 20% in Madrid and 10% and 14% in Boston. Design Advisor showed the largest MPE of the studied tools for the cooling load estimation, with a 37% difference in the monthly average value for the Madrid case study. In conclusion, yearly energy totals were predicted with less error, which is a reasonable outcome since savings are not typically reported weekly or daily.

*4.2. Daily and Hourly Analysis*

To evaluate daily and hourly predictions, general-purpose or specific-purpose advanced simulation tools were analyzed. For example, TRNSYS is an excellent option to model new technologies and analyze specific building parts. However, it was not used in this article due to its complex whole-building simulation. On the other hand, IDA ICE showed the best compatibility with Revit, and the export of the 3D model did not need extra work, so it was used in this study to represent this kind of tool. Specific-purpose tools were represented by the engine EnergyPlus with a Design-Builder interface. The larger the dataset, the more robust the evaluation of the deviations is [48]. Therefore, the deviations were assessed using hourly values for heating and cooling. In addition, the average value of all tools was used as a benchmark for each parameter. ASHRAE Guideline 14-2014 and the International Performance Measurement and Verification Protocol (IPMVP) [49,50] establish a method for measuring the precision of energy models in buildings. In addition, these documents suggest several thresholds for monthly or hourly calibration for the Normalized Root Mean Square Error (NRMSE) [51]. This analysis was completed utilizing monthly and hourly data for heating and cooling. Figure 10 shows the hourly energy consumption for heating and cooling in Madrid for four months to illustrate the characteristic values in all the seasons. The hourly energy consumption showed similar patterns for both tools in July, with a 77-kWh difference between the monthly cooling energy in IDA ICE and Design-Builder. The former showed a peak value of 17.95 kW, and the latter, 21.92 kW. The after-hours energy consumption in the IDA ICE simulation compensated for the highest peak value of the Design-Builder cooling energy. The same pattern is observed in January, with peak values of 21.23 kW in Design-Builder and 10.11 kW in IDA ICE. The monthly heating energy consumption was 2030 kWh and 2237 kWh for IDA ICE and Design-Builder simulations, respectively.

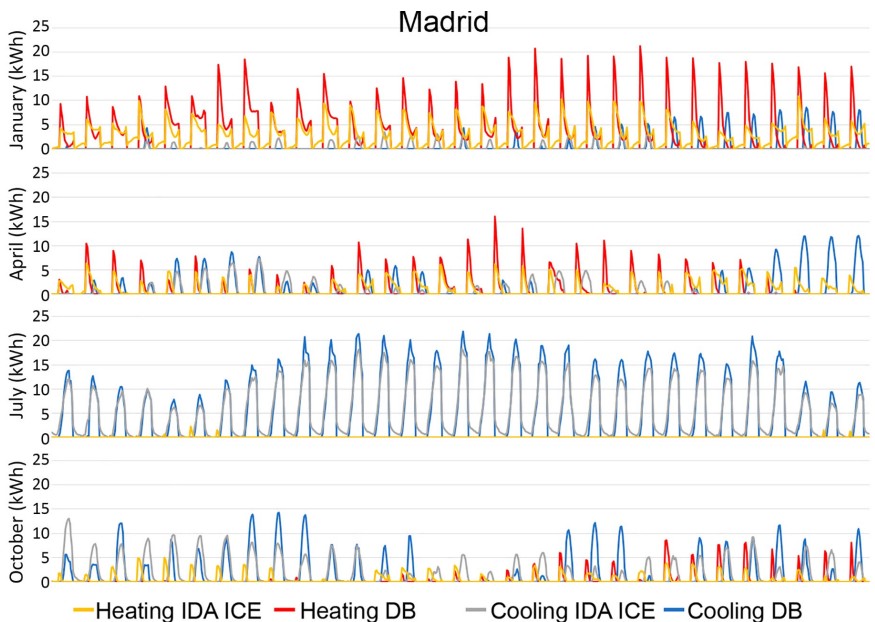

**Figure 10.** Hourly energy demand of the Madrid case study simulated with Design-Builder and IDA ICE.

The hourly energy loads were different for both software tools in April and July. The deviations were due to the merged impact of the various sky models and the calibration of the heating and cooling systems, which allows reaching the same energy load, but with a different hourly distribution. Figure 11 shows the Boston energy consumption plots for heating and cooling for four months. The monthly energy consumption in July was 3486 kWh for IDA ICE and 4124 kWh for Design-Builder, with peaks of 16.26 kW and 25.05 kW, respectively. The same deviation can be found in January with an energy consumption of 4352 kWh and 5329 kWh for IDA ICE and Design-Builder. The behavior in April and October was different, and IDA ICE showed higher values for both heating and cooling. The energy consumption was 1688 kWh and 1212 kWh in April for IDA ICE and Design-Builder, respectively.

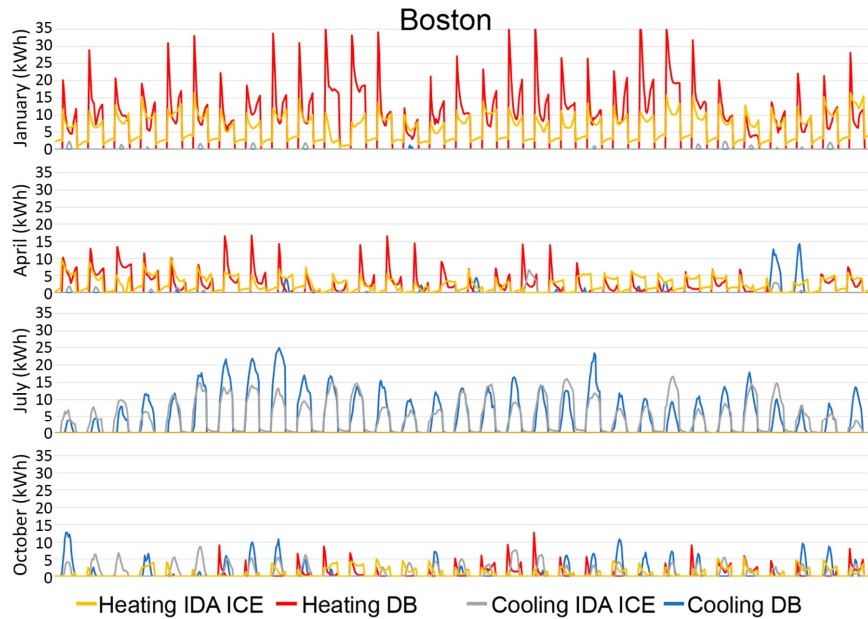

**Figure 11.** Hourly energy demand of the Boston case study simulated with Design-Builder and IDA ICE.

Figure 12 shows the detailed hourly results of energy consumption for each tool over 48 h in Madrid and Boston. The Design-Builder heating plot showed 20.5 kWh and 34.9 kWh peak values in Madrid and Boston, respectively. The IDA ICE pattern was steadier and showed energy consumption when the building was not used from 8:00 p.m. to 6:00 a.m. Two parameters could explain the deviations between both tools: the outdoor temperature and the solar irradiance, represented by the solar gains through windows. The latter showed similar values for both tools in summer and winter. The former showed important deviations.

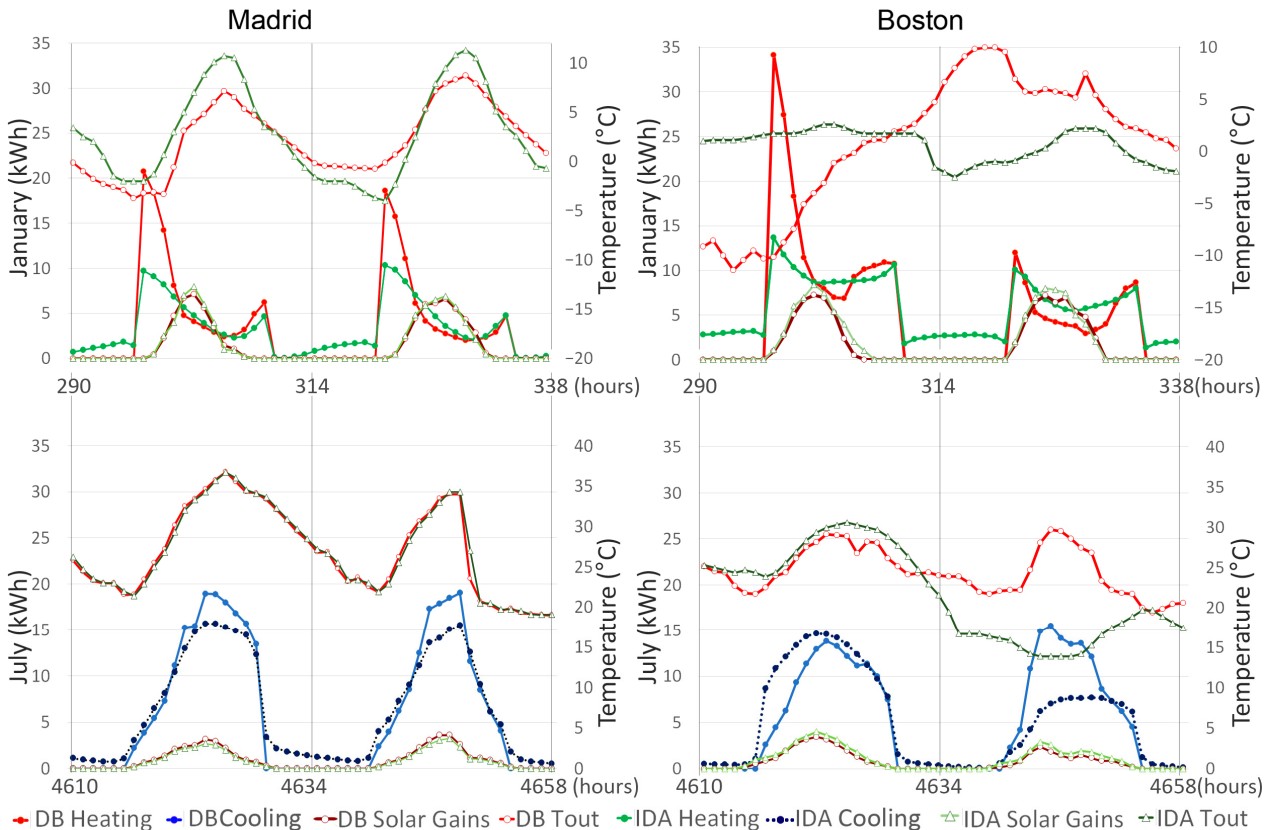

**Figure 12.** Hourly energy demand of the case studies simulated with Design-Builder and IDA ICE for 48 h in January and July.

The default method of infiltration modeling by IDA ICE is the wind-driven flow based on leak sizes and wind pressure. With this method, leaks are automatically introduced in each external wall, and then they are distributed to each zone. In Design-Builder, the default infiltration approach establishes a set of nodes connected by airflow elements. For a more accurate simulation, a relationship between airflow and pressure must be specified for each element, so that effect was not considered for this work. With the IDA ICE model, the air infiltration flow rate can lead to higher energy consumption in winter, so the heating loads observed could be attributed to the constant value set in the simulations. The default assumptions for thermal mass explained the deviations at the starting time of the heating system in Boston and Madrid. Thermal mass does not affect the geometry but is essential to heat transfer calculations, for example, furniture within the space, particularly in large spaces, or conventional wall structures with equivalent U-values and different thermal masses.

ASHRAE Guideline 14-2014 [49] describes a method for validating a building model against measurements and suggests the limits for the root mean square error (RMSE) and

the normalized root mean square error (NRMSE), shown in Equations (4) and (5), using the average as a normalization means, to verify the accuracy of the models.

$$RMSE = \sqrt{\frac{\sum_{i=1}^{n}|S_{Si} - A_{Ri}|}{n}}, \tag{4}$$

$$NRMSE = \frac{1}{nm}\sqrt{\frac{\sum_{i=1}^{n}|S_{Si} - A_{Ri}|}{n}}, \tag{5}$$

where *nm* is a normalization mean defined by Equation (6). As per ASHRAE recommendations, the normalization mean was the average of reference values $A_{Ri}$ considering only numbers higher than zero, $n_{A_{Ri}}$.

$$nm\,(av > 0) = \frac{\sum_{i=1}^{n} A_{Ri}}{n_{A_{Ri}} > 0}. \tag{6}$$

Table 5 shows the values of RMSE and NRMSE for the heating and cooling loads. Those indices were calculated on an hourly basis for each month.

**Table 5.** Mean error and mean percentage error for yearly results from IDA ICE, Design-Builder, and Design Advisor for both case studies.

| Month | 1 | 2 | 3 | 4 | 5 | 6 | 7 | 8 | 9 | 10 | 11 | 12 |
|---|---|---|---|---|---|---|---|---|---|---|---|---|
| RMSE$_{(heat)}$ | 2.9 | 1.9 | 1.7 | 1.3 | 0.4 | 0.2 | 0.0 | 0.0 | 0.2 | 0.8 | 2.1 | 2.7 |
| $n_{A_{Ri}} > 0$ $_{(heat)}$ | 745 | 672 | 744 | 709 | 506 | 221 | 17 | 53 | 236 | 393 | 678 | 742 |
| NRMSE$_{(heat)}$ | 43.5 | 33.6 | 41.8 | 62.2 | 61.5 | 95.2 | 78.3 | 80.2 | 98.7 | 61.7 | 67.2 | 55.3 |
| RMSE$_{(cool)}$ | 0.2 | 0.2 | 0.3 | 0.8 | 0.4 | 2.4 | 1.8 | 2.5 | 0.8 | 1.1 | 0.6 | 0.3 |
| $n_{A_{Ri}} > 0$ $_{(cool)}$ | 87 | 95 | 117 | 123 | 259 | 406 | 690 | 505 | 586 | 287 | 112 | 59 |
| NRMSE$_{(cool)}$ | 49.5 | 40.6 | 46.3 | 60.1 | 15.9 | 35.5 | 34.9 | 41.0 | 25.0 | 46.7 | 37.0 | 24.0 |

The calibration criteria given by these standards is 30% for the hourly NRMSE, which was not met for heating loads. The problem with this analysis was that the average heating and cooling energy consumption was close to zero in periods when the building did not need any heating or cooling power. Thus, the normalization factor led to a lower NRMSE for heating in the winter months, even though the RMSE was higher than the values in the summer months. The analysis of cooling loads showed acceptable values, close to 30%, from May to September.

## 5. Conclusions

This article's methodology relied upon analyzing a simulated data set and applying cross-validation to compare energy consumption predictions among different tools on the same characteristic office cell. By varying the unit of energy prediction (daily, monthly, and yearly energy use), building designers and managers can understand how errors change with different reporting intervals.

Design-Builder and IDA ICE did not outperform a basic simulation tool, such as Design Advisor, in the yearly analysis. In this regard, the mean absolute error was minor for a longer prediction time. This outcome was positive because savings are stated yearly in most building codes and energy efficiency standards. Thus, for a preliminary analysis of different design options, Design Advisor is recommended when the final geometry of the project still needs to be completed. After several iterations, the mean percentage error (MPE) was below 6% for the yearly energy loads calculated by all tools. The monthly MPE ranged between 12% and 20% in Madrid and 10% and 14% in Boston.

The normalized root mean square error (NRMSE) was used for larger datasets, such as hourly values, per the recommendation of ASHRAE Guideline 14-2014. After this analysis, most of the monthly deviations for heating loads were above the 30% threshold set by the ASHRAE standard. Only the cooling load analysis showed acceptable values, close to 30%,

from May to September. The systematic deviation between the energy consumption of the tools was primarily a function of the outdoor temperatures for both weather files for each simulation tool in both locations and simulated periods. Other reasons for the deviations were the different mathematical models used to simulate the air infiltration flow rate and the influence of the thermal mass between IDA ICE and EnergyPlus.

Further studies must continue to test the proposed methodology in buildings with increased complexity in heating, ventilation, and air conditioning systems and compare with actual values from monitoring systems. Unqualified users must not use these software tools as "black boxes". Standard-of-practice requires that firms have highly trained, professional users of such tools on staff or outsource such detailed energy studies to one of the many qualified building energy simulation consultants. Doing otherwise does not meet the client's needs and faces design firms with significant liability.

**Author Contributions:** Conceptualization, F.D.A.G. and B.M.S.; methodology, F.D.A.G.; software validation, B.M.S., F.D.A.G. and M.J.M.B.; formal analysis, B.M.S. and F.D.A.G.; investigation, F.D.A.G.; resources, B.M.S.; data curation, F.D.A.G., B.M.S. and M.J.M.B.; writing—original draft preparation, F.D.A.G.; writing—review and editing, M.J.M.B. and B.M.S.; visualization, F.D.A.G.; funding acquisition, F.D.A.G. All authors have read and agreed to the published version of the manuscript.

**Funding:** This research received no external funding.

**Institutional Review Board Statement:** Not applicable.

**Informed Consent Statement:** Not applicable.

**Data Availability Statement:** Not applicable.

**Acknowledgments:** This work was supported by Keene State College Faculty Development Grant program.

**Conflicts of Interest:** The authors declare no conflict of interest.

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
