# Peer review of "Assessment of Building Energy Simulation Tools to Predict Heating and Cooling Energy Consumption at Early Design Stages"

_sustainability, doi:10.3390/su15031920_

Round 1

Reviewer 1 Report

The study named "Assessment of Building Energy Simulation tools to predict Heating and Cooling Energy consumption at early design stages." is remarkable in terms of its subject. However, it includes some of the handicaps listed below.

1) Introduction section is quite weak. It should be designed and developed well.

2) Related works should be given in another section. And it should include the more detailed evaluation of the studies.

3) I recommend to write the contributions of the study as detailed.

4) The conclusion part of the study is quite weak and it should be developed well.

5) Authors should explain other methodologies used in more detail in the relevant section. Thus, the structures of other methods used can be better understood. After this stage, they can also include the parameters in the relevant methodologies. Thus, the difference of the algorithm from other algorithms will be shown better.

6) The figure quality in the study is too low. These figures need to be rearranged and given in high resolution.

Reviewer 2 Report

1.     The background part of the abstract is too long, it should focus on the work of the authors. The abstract part fails to show the highlights of this manuscript, so it is recommended to rewrite the abstract.

2.     In the first part, the authors introduce the existing simulation software in the research market, but does not compare them. What are the characteristics of these software? What are the main differences between them? It is recommended to add a more detailed introduction.

3.     The authors mentioned in the abstract is not to study specific buildings, but to analyze this case and use it for more architectural research. We all know that different building characteristics will have different forms of energy consumption. In lines 232-236, I want to know the basis of the rectangular office building selected by the authors as a typical building in the manuscript? How can it represent the characteristics of other buildings?

4.     In the 462 line, the picture is generally followed by the above introduction, and the data in the picture is analyzed and described below the picture. Other pictures in the manuscript also have the same modification suggestions.

5.     All the data generated in this paper are calculated by software. I haven't seen the author compare the simulated calculation data with the actual data. Even in the conclusion, there are no other results of analysis and comparison. The author should at least give in the conclusion which software has its unique computing advantages under specific conditions, and how to choose the appropriate computing software according to the building type, and so on. I think the description of the authors in the conclusion is too long, so it should be simplified in the conclusion part, and only the main conclusions should be elaborated.

6.     The overall length of the article is very long, but most of the content is about the introduction of software and software operation methods. All data are directly generated by the software, without the author's own analysis. The manuscript is more like a case calculation than a scientific research paper.

Reviewer 3 Report

This manuscript describes, from an architectural-user's point-of-view, operation of and results from a few common building energy simulation programs.  There are many such publications, so this work is not unique.  However, as the programs evolve periodic review is important.

This manuscript still needs some work, however.

Of first importance is to describe/clarify early in the manuscript the difference between the "user-hostile" calculation cores and the "user-friendly" shells, often prepared by different programmers and organizations.  For example, the originally U.S. government-funded cores to DOE2 (essentially an advanced CLTD method; considered obsolete; is _not_ EnergyPlus as you stated early), BLAST (transfer function method, also now considered obsolete), TRNSYS (a solver of differential equations; applied originally for transient simulations of active solar-thermal energy systems), and EnergyPlus ("state-of-the-art" transient Heat Balance Method solver for building energy simulations).  Basic shells were also typically provided by the cores' developers, normally only for use by developers and researchers, not building designers.

Then introduce the much more user-friendly shells, ending with the ones you compare.

 eQUEST, despite using DOE2, is still widely used due to its ease and being free.

Some have developed shells and additional modules, in FORTRAN, for TRNSYS, but most do not consider using a TRNSYS core for whole-building simulations -- instead it is best applied to detailed study of an individual component or mechanical system (so "research").

There are many shells that use the EnergyPlus core -- state clearly which of your four final shells use EP.  If not, what are their calculation cores, and the basic calculation method used?

If two or more of your final shells use the same core, and if using the same modern version of such core, state clearly why there would be differences in the results -- e.g., user experience/error, shell's default values, limitations on each shell's input, ...

The various papers from "The Great Energy Predictor Shootouts" may be helpful for determining these differences.

Some specifics:

Throughout, remove "commercial"/biased statements or tone.  E.g., L104's "most reliable" (without citation), L220's "industry leading," ...

Section 2.4 needs better flow/continuity of ideas, including use of paragraphs.

L232+234: Not "remediation," air exchange.  Not "permeability," infiltration or air leakage.

L328: A fixed ACH sounds like a default value instead.  Most likely the ventilation rate is user-adjustable too; if not, that alone would make that (early?) version of the software inappropriate to use for building energy modelling.

In Conclusions, end with the expected warning statement along the lines of "these programs are, of course, just tools that must not be used as 'black boxes' by unqualified users.  Standards-of-practice require that firms have highly qualified, experienced users of such tools on staff, or to contract such detailed energy studies to one of the many qualified building energy simulation consultants.  Doing otherwise does not meet our clients' needs and opens design-firms to significant liability."

Reviewer 4 Report

The manuscript entitled “ Assessment of Building Energy Simulation tools to predict 2 Heating and Cooling Energy consumption at early design 3 stages.”  is interesting. Some recommendations are as followed to further improve the content of the manuscript.

i)                   Line 99- Since this study already completed, “will study” shall change to “studied”. Same goes to line 103.

ii)                  Line 426- many iterations are subjective. Kindly provide the number of iterations.

iii)               Author shall include citation for Equation (1) if it is not derived by authors.

iv)               All figures shall be revised. The current form is not clear. Would suggest increasing the dpi and quality of figures. **Especially Figure 12.

v)                  Authors shall highlight and summarise the findings from Table 2 -table 5.

vi)               Discussions on page 16 and page 17 are limited. Authors are suggested to discuss and consider including critical analysis. Discussion based on the figure itself is not sufficient.

vii)             Line 169- abbreviation of cfd shall be defined when first introduce it (line 161 and not supposed to be at line 169).

viii)           Recent citations are needed at the back to support this statement “In addition, it can calculate the impact of supply air distribution on temperature and velocity distribution within a room with CFD.”. Some publications are suggested as follow:

 https://doi.org/10.1016/j.buildenv.2022.109489

https://doi.org/10.1007/s10973-022-11466-6

https://doi.org/10.1007/s11356-022-23407-9

ix)               Line 191- “IDA ICE is a professional software”. I think this statement is not appropriate. Many software also considered as professional software. Would suggest authors to rework on this sentence. Maybe provide brief introduction “what is IDA ICE” software would be more appropriate.

Round 2

Reviewer 1 Report

The authors replied my comments well  but the figure quality in the study is quite low. Authors should check the settings of the "MS Word".These figures need to be rearranged and given in high resolution.

Author Response

Thank you for the advice.
We have changed the advanced settings of MSWord to insert figures with a "High Fidelity" resolution. Then, we improved the figures' resolution and inserted them again into the text.
We hope the quality of the figures is acceptable now.

Reviewer 2 Report

In this revision, the authors added the description of building selection in the calculation case, changed the order of some pictures in the manuscript, deleted the description of building energy consumption software, and added numerical content in the abstract and conclusion, which enhanced the preciseness of the article and significantly improved the quality of the manuscript. This modification basically solved the problem I mentioned. However, the conclusion is still too long, it is more like a discussion than a conclusion, so it is suggested to further delete it and only keep the main conclusion.

Author Response

Thank you for your revision and advice.
We have cut some paragraphs from the Conclusion section and pasted them into the Discussion section. The Conclusions are focused on the main findings of the article now, according to other reviewers' suggestions.

Reviewer 3 Report

Much improved!  Good job, with one minor exception:

Lawrence Berkeley National Laboratory {note their spelling of Lawrence}.  :-)

Author Response

Thank you for the advice.

We have changed Laurence by Lawrence, which is the correct spelling.